# A Retrospective Longitudinal Analysis of Mental Health Admissions: Measuring the Fallout of the Pandemic

**DOI:** 10.3390/ijerph20021194

**Published:** 2023-01-10

**Authors:** Sean Warwicker, Denise Sant, Adrian Richard, Jake Cutajar, Annalise Bellizzi, Gertrude Micallef, Daniel Refalo, Liberato Camilleri, Anton Grech

**Affiliations:** 1Mount Carmel Hospital, ATD 9033 H’Attard, Malta; 2Mater Dei Hospital, MSD 2090 Msida, Malta; 3Department of Psychiatry, Faculty of Medicine and Surgery, University of Malta, MSD 2080 Msida, Malta

**Keywords:** mental health, COVID-19, healthcare access, healthcare inequality

## Abstract

Background: In this research article, we review the infrequently considered long-term impact of the pandemic on inpatient mental health, by reviewing the clinical parameters of all psychiatric admissions to Mount Carmel Hospital, our region’s main psychiatric healthcare facility, from 2019–2021. Methods: 4292 patients were admitted during the research period of this retrospective longitudinal analysis. Taking 2019 as the pre-COVID reference year, we compared mean monthly admissions from 2020 and 2021, looking at patient demographics, status under the Mental Health Act, diagnosis, and self-injurious behaviour. Results: While the pandemic was reflected in a moderate increase in mean monthly presentations with suicidal ideation and suicidal self-injury, presentations in 2020 otherwise remained largely stable. This contrasted with a surge in presentations in 2021 with mood disorders, schizophrenia, anxiety, personality disorders, and autism spectrum disorders. Furthermore, presentations involving self-injurious behaviour continued to grow. Involuntary admissions also increased significantly in 2021. Conclusions: This paper highlights the pernicious long-term impact of the pandemic on mental health presentations, demonstrated by an increase in hospital admissions and more serious presentations. These findings should be considered in the guidance for responses to any future pandemic, giving attention to the evidence of the impact of restrictive measures on mental health.

## 1. Introduction

The repercussions of COVID-19 on mental health and the public health measures that were established during the first year of the pandemic have been well studied, although notably, the findings of these studies have varied, reflecting the chaotic process of adjustment that swept the world in response to the virus. Less attention has been paid to the longer-term effects on psychiatry now that COVID-19 has become an established part of everyday life and, critically, the consequences of lockdowns and other related restrictive healthcare strategies have been established.

While lockdown periods were generally associated with commensurate decreases in psychiatric admissions [1,2,3,4], the magnitude of this change in service utilization was not universal [5]. Indeed, in the UK, admissions related to mental health were observed to increase in 2020 compared to 2019, in some centres [6]. Policies guiding the control of the pandemic were diverse and are likely to have influenced the incipient disruption that was experienced in psychiatric admission rates. Greater rates of anxiety [7,8], depression [8,9], and suicidal ideation (SI) [1,7] were observed in some centres, although this contrasted with findings of a fall in presentations with non-suicidal self-injury (NSSI) during lockdown in the UK, observed most prominently in women [10]. This was despite evidence from other sources demonstrating that the frequency of SI and NSSI reported during the first month of lockdown in the UK followed pre-pandemic patterns, and included a greater prevalence among women and ethnic minorities [11].

While avoiding the strict lockdowns that were seen in other parts of the world, the public health strategy which guided the response to the pandemic in Malta was nonetheless comprised of a myriad of restrictive measures. The first cases of COVID-19 were confirmed in Malta on the 7th of March 2020, in a family returning to the country from Italy [12]. Initial rules mandating 14 days of self-quarantine on return from only the most affected regions of Italy were expanded to encompass other highly impacted European countries at the time, with outright prospective bans on sea and air travel to these regions announced on the 11th of March [13]. The first major local restrictions were imposed the following day, with the closures of educational institutions and care centres for children and the elderly. Religious gatherings were discontinued, and all political activities stopped [14].

As viral cases continued to rise, further measures which were taken that month included extending the closure of schools until the end of the academic year [15], the temporary discontinuation of all non-essential services [16], and a ban on all organized group gatherings [17]. Violations of these policies were initially to be met with fines of €3000, which then rose to €10,000 as the country grappled with a worsening situation [18]. Mandatory 14-day quarantine periods were rigorously enforced.

Numbers of COVID-19 cases eased over the summer, then Malta experienced a resurgence towards the end of the year. From October, regulations on the wearing of masks were strengthened and masks became compulsory in all schools, workplaces, and outdoor public areas. Strict curfews were imposed on recreational centres, and restrictions on group gatherings were maintained. While these actions provided an interval of respite, viral cases grew markedly in March of the following year. Some curtailments on retail sectors, schools, and workplaces had been partially lifted by this stage, but these were brought back into force following this spike in cases, and group gatherings were limited to four individuals [19].

The eventual sustained lifting of public restrictions was contingent upon individual uptake of an approved COVID-19 vaccination [20]. In May 2021, Malta became the first EU nation to make the vaccine available nationwide. Quarantines for those who had been vaccinated but then exposed to an active COVID-19 case were reduced to 7 days, public restrictions were again gradually eased, and the overall outlook improved during the warmer summer months [21]. Malta joined many of its European counterparts in issuing “vaccine certificates”, granting greater freedoms to those who were fully vaccinated, such as access to certain retail and recreation sectors [20]. Further vaccine certificates were issued following the uptake of a COVID-19 booster injection within the context of yet another spike in cases towards the end of the year, as the Omicron variant became the predominant viral strain. While some restrictions were reinstated, they were neither so broad nor severe as had been experienced previously [22,23].

The conflicting nature of the conclusions in the flurry of international literature published in the opening months of the pandemic was probably influenced by the lack of large-scale observational studies, which provided only brief snapshots of a nebulous period in time [8]. Furthermore, there has been a notable paucity of data on the impact of the pandemic on inpatient psychiatry. In this study, our primary aim was to evaluate the long-term impact of the pandemic on psychiatric hospital admissions. We considered data for admissions in the first two years of the pandemic (2020 and 2021) to Malta’s Mount Carmel Hospital, the region’s major psychiatric inpatient facility, and compared them to data for 2019, which served as the pre-pandemic reference year.

The secondary aim of our study was to link the observed changes in psychiatric hospitalizations to Malta’s pandemic timeline. Over the course of the pandemic, Google used the same accumulated and anonymized data that it gathers for its mapping features to chart movement of people over time and area [24]. Using this information, Community Mobility Reports for individual countries were generated with the intention of guiding public health policy and planning. These reports provided valuable insight into the impact of COVID-19 restrictions, and the data remains publicly available for research purposes. The data are presented as daily percentage changes from baseline across six geographical domains. Data for Malta was first shared in February 2020 and continued to be recorded into 2022, providing a useful objective parameter against which we were able to correlate hospitalizations. These mobility trends are charted against mental health hospitalizations in Figure 1 and are explored further in this paper.

## 2. Materials and Methods

All psychiatric inpatient admissions to Mount Carmel Hospital in the years 2019–2021 were reviewed in this retrospective longitudinal analysis. The Shapiro–Wilkes test was applied to assess for normality of the distribution of monthly admission numbers across the compared categories. As the vast majority of our data was normally distributed, Student’s paired T test was applied to compare mean monthly data for 2019 (the pre-COVID reference year) with 2020 and 2021, and to compare 2020 with 2021. In order to avoid type II error, this method of interpretation was selected in favour of a comparison of proportions for each year, because increases in admissions across several diagnostic blocks led to an absolute increase in admissions in 2021. Research approval was gained from the Institutional Review Board of Mount Carmel Hospital.

For the purposes of our research, the inclusion criteria were patients who were either newly admitted or re-admitted (following a prior formal discharge process) in view of psychiatric illness. During this 3-year period, 5760 patients presented to the hospital. Of these patients, 415 returned to the hospital from leave (meaning that these patients were not formally discharged by their responsible psychiatric care provider and were therefore excluded from this study); 431 returned following transfer from Mount Carmel to a general medical hospital for non-psychiatric indications, and were thus also excluded; 50 were excluded for miscellaneous reasons (including double-entry or admission in which no clear psychiatric disorder was specified); and 572 were excluded due to insufficient clinical records. A total of 4292 psychiatric admissions were therefore recorded during this period. Over the timeline of our study, 1348 admissions were recorded in 2019, 1378 in 2020, and 1566 in 2021.

Criteria assessed included patient gender, age group, and nationality. We also assessed Mental Health Act (MHA) status on admission (patients were broadly divided into either voluntary or involuntary status, the involuntary patients were only those requiring admission under a branch of the MHA, not those who were admitted voluntarily but then later detained under the MHA), clinical diagnosis (according to the ten ICD-10 mental health diagnostic blocks); and the respective proportions of presentations with NSSI, SI, or suicidal self-injury (SSI). It was noted that all admissions in the F50–59 diagnostic block were attributable to eating disorders, while those in the F80–89 block were attributable to diagnosis of an autism spectrum disorder (ASD).

Some patients may have been assigned more than one psychiatric diagnosis following admission, but only those considered as contributing directly to their presentation were recorded for this study. For example, for a patient with a previous psychiatric history of mood disorder admitted during the study period with first-episode psychosis with no major affective symptoms, only the diagnostic block for the psychosis would be recorded. Data were gathered using electronic case summary (ECS) records, equivalent to patient discharge letters and electronic Emergency Department (ED) records in cases where the patient’s first presentation was to the ED.

For the second part of our analysis, mean monthly percentage changes from baseline were taken from the Google Community Mobility Reports. These reports gathered data from six domains: retail and recreation, grocery and pharmacy, parks, transit, workplaces, and residences. Pearson’s correlation was calculated for these values and the total numbers of hospitalizations per month for the 23-month period between February 2020 and December 2021. Because daily mobility reports for Malta were first made available on the 15th of February 2020, the mean value for only the last two weeks of that month were correlated with the respective monthly admissions. These reports measured how populations moved over time throughout the pandemic, and their results were taken as proxy values for the weight of restrictions and attitudes towards COVID at different points in time.

Statistical analyses were performed using IBM Statistical Package for Social Sciences (SPSS) Version 26 (IBM, Armonk, NY, USA) for Windows.

## 3. Results

The mean monthly admission data regarding clinical diagnosis are presented in Table 1. The highest admission rates in 2019 were for substance use disorders, mood disorders, schizophrenia, schizotypal and delusional disorders, anxiety disorders, and personality disorders, respectively. As can be observed in the demographic data recorded in Table 2, numbers of voluntary admissions exceeded those of involuntary, and males were much more likely to be admitted than females over the year. On a mean monthly basis, 5.6 (SD 2.54) patients presented with NSSI, 23.2 (SD 3.79) with SI, and 7.8 (SD 2.08) with SSI (Table 3).

There was a shift in patient parameters in the first year of the pandemic, although it was noted that much of the variation was not statistically significant. The total number of admissions for the year was similar to that for 2019, though an increase in non-Maltese admissions may have reflected a disproportionate socio-economic impact of the pandemic (22.7/month vs. 28.7/month, *p* = 0.005). There were no significant differences in admission rates based on sex or MHA status, and the only significant change in terms of psychiatric diagnosis was a fall in childhood behavioural and emotional disorders (2.8/month vs. 0.8/month, *p* = 0.002). Presentations with SI increased (23.2/month vs. 36.1/month, *p* = 0.004), and there was a borderline significant increase in presentations with SSI (7.8/month vs. 11.3/month, *p* = 0.05).

Strikingly, there were much greater and broader changes in 2021, the second year following the onset of the pandemic. A surge in total admissions for the year was due to significant increases in admissions for mood disorders (27.2/month vs. 42.5/month, *p* = <0.001), schizophrenia, schizotypal and delusional disorders (17.9/month vs. 23.8/month, *p* = 0.002), personality disorders (6.9/month vs. 12.6/month, *p* = 0.012), anxiety disorders (15.8/month vs. 20.3/month, *p* = 0.011), and ASD (1.5/month vs. 3.0/month, *p* = 0.032). The increases in mood and anxiety disorders each represented a significant rise from 2020 levels, as did the rise from 2020 observed in presentations with behavioural and emotional disorders of childhood, which returned to pre-pandemic levels. The difference in sex distribution was accounted for by a significant increase in female admissions compared to the two previous years, and there was a marked increase in mean monthly involuntary admissions (31.5/month vs. 46.6/month, *p* = 0.001). Non-Maltese admissions remained stable in comparison with 2020 levels, while there was a significant increase in Maltese admissions compared with pre-pandemic levels (89.7/month vs. 102.8/month, *p* = 0.028).

The rate of presentations with self-injurious behavior only grew in 2021. Presentations with NSSI grew to a level significantly greater than was observed in 2019 (5.6/month vs. 9.0/month, *p* = 0.03), while presentations with SI (23.2/month vs. 44.8/month, *p* = <0.001) or SSI (7.8/month vs. 13.3/month, *p* = 0.003) increased at rates exceeding those observed in 2020. The only significant increase observed in the comparison of 2021 with 2020 was for SI.

In the results of our analysis of correlation between total monthly admissions and community mobility, there were varying degrees of significant correlation, the majority of which were strong. As can be observed in Table 4, the strongest positive correlations were with mobility in parks, transit, retail, and recreation. The only area of negative correlation was with residential mobility. Figure 1 provides a graphical representation of how hospitalizations waxed and waned throughout the pandemic, indicating the clear temporal relationship with the various mobility trends. The mobility trends themselves reflect the varying impacts of local governmental policies, fluctuations in COVID-19 case numbers, and changing views towards the pandemic. Scatter plots in Figure 2 provide a visual representation of the varied relationships between monthly admissions and mobility data.

## 4. Discussion

With regards to the primary aim of our study, the findings from the first pandemic year are in many ways consistent with the divergent nature of the literature which emerged during the period: several small and generally statistically insignificant changes, albeit with early indications of presentations with deterioration in mental health, as highlighted by an increase in presentations with SI or SSI. Taken in the context of the results for 2021, these findings paint a picture of transition that can only be appreciated from a broader perspective.

While we noted a drop in admissions in March and April 2020, coinciding with the region’s first recorded COVID-19 cases [12], the total number of admissions for the year increased slightly compared to 2019. The continued growth in psychiatric admissions in 2021 beyond pre-pandemic levels was more notable than the short-terms fluctuations in 2020 (1566 vs. 1348). Longer-term observational data are beginning to highlight this chronic effect [25], building on the knowledge provided by earlier short-term studies that focused more on reduction [1,2,3,4,10,26,27] and subsequent rebound [6,25,27] in patients presenting with mental health problems. The interruptions to healthcare provision during the pandemic, consequences of prioritization in resource deployment and of patient avoidance [2,6,25,26,27,28], no doubt contributed to this upward trend, while the challenges posed by inpatient psychiatry have proven particularly complex [29].

Mood and anxiety disorders have to date featured extensively in the discussion regarding the impact of the pandemic on psychological wellbeing. Our findings confirm that not only did admissions across these two diagnostic blocks continue to grow during the second year of the pandemic, but also that more patients were admitted with psychotic disorders, personality disorders, and ASD, an upswing that was not yet perceptible in 2020. When these results are taken in the context of consecutive increases in involuntary admissions observed for the two years of the pandemic, and the year-on-year increases in presentations with NSSI, SI, or SSI, they suggest that psychiatric presentations were more frequent in 2021, and were also more severe.

It should be emphasised that this study was conducted in an area that was able to avoid the sweeping lockdown policies enacted elsewhere during the peaks of COVID-19, but which nonetheless employed a considerable level of restrictions. One year into the pandemic, 55% of the population felt that their personal income had either already been affected or would be in future. Uncertainty, frustration, and fear were among the most common themes to emerge from EU-wide survey data [30]. In spite of the implementation of healthcare outreach programmes such as telemedicine, the drop in face-to-face visits was precipitous both locally and across Europe [31]. Indeed, while the adoption of telemedicine is often heralded as one of the success stories of the pandemic, there is a strong case to be made that it aggravated existing socio-economic inequalities in some respects [32,33]. Older age groups are less likely to engage in digital healthcare services, representing a major barrier to care [34]. Our results suggest that this factor could have played an important role, with the 60–69 age group experiencing the most significant proportionate increase in admission rates in 2021 vs the pre-pandemic period, with little difference noted for those aged under 30. Furthermore, unequal access to care probably played a part in the rise in international admissions recorded in 2020 and 2021, which occurred despite a downward trend in refugees and migrants arriving in the country during the study period [35], with this population having a substantial mental health burden of their own.

The available evidence reveals that outcomes are comparable between telepsychiatry and in-person assessment in terms of emergency assessment [36], short-term clinical results [37], patient and provider perspectives [38], and cost-effectiveness [37,38]. This further suggests that the accessibility of telepsychiatry may have been a contributory factor in the changes observed in inpatient psychiatry. Further compounding these issues are inherent risks of the virus itself, including diverse neuropsychiatric sequelae triggered by direct viral invasion or via potential immune-mediated effects that ripple through the central nervous system [39,40]. Upregulation of pro-inflammatory cytokines and disruption of the hypothalamic–pituitary–adrenal (HPA) axis in COVID-19 infection have been linked to depression, anxiety, and psychosis [40], all areas in which we observed significant increases in hospital admissions.

The results of the correlational analysis of our study, taken together with the decrease in presentations observed over the course of the pandemic, further implicate the roles of healthcare access and equality. The mobility reports provided by Google provide a fascinating view of how the pandemic and the public health strategies that followed were felt by the Maltese population. The strength and significance of the correlation that we observed between admissions and the mobility data suggests the likelihood that, irrespective of the causes of variation in mobility, psychiatric presentations warranting hospitalization were delayed. While it may be argued that outreach programmes and telehealth might have adequately supported this population at times when the strongest restrictions were being enforced and anxiety about COVID-19 was at its highest, the long-term deterioration apparent in 2021 would indicate otherwise.

Over the span of the pandemic, admissions increased when people were spending more time in public places and fell when they were more restricted to their homes. While it is impossible to calculate the degrees to which different factors contributed to changes in mobility, we interpreted mobility at different timepoints to be a product of restrictions (such as school closures, curfews, quarantines, and limits on group gatherings), attitudes towards COVID-19 case numbers, attitudes towards the perceived threat from the virus (which would vary based on media reporting, viral strain, and vaccine uptake), and the indirect effects of the pandemic such as job losses and financial constraints. There are probably a multitude of other factors which could also be implicated here, such as altruistic self-imposed isolation from elderly or vulnerable loved ones that many opted for, especially at the start of the pandemic.

It is interesting to observe that workplace mobility remained below baseline throughout the pandemic period, and that the converse can be seen for residential mobility, reflecting the shift towards working from home which took place in Malta. Furthermore, it is unsurprising that the smallest fluctuations in mobility were observed for groceries and pharmacies, sectors which would have been heavily used by the population as viral cases trended upwards. The very strong correlation with parks was probably a result of the fact that more visits to these areas would no doubt have occurred at times of an improved sense of safety and security. Mobility in this aspect was greatest during the warmer summer months, which were periods of decreasing case numbers and reduced restrictive measures. Healthcare access may certainly have been better facilitated.

The worsening picture of psychiatric presentations that we recorded was no doubt a result of this constellation of direct and indirect factors that ebbed and flowed throughout the pandemic, predisposing vulnerable populations to long-term adverse outcomes. It would be remiss not to consider at this point the impact of school closures on children’s and adolescents’ mental health. We observed that in 2020 there were significant drop in admissions for behavioural and emotional disorders of childhood onset and in the under-18 age group overall. Interruptions to educational services not only impeded access to important resources including an environment providing daytime structure [41], but also withdrew a critical setting where young people might first present with mental health difficulties. This may be particularly true when young people present with externalizing behaviours, such as aggression, impulsivity, or hyperactivity. School closures were yet another area where existing socio-economic disparities were exacerbated. Interestingly, mental health outcomes have been shown to differ between children who attended school remotely and those who attended in person [42], compared with the remote vs in-person healthcare delivery format.

2021 admission rates for disorders of psychological development, especially ASD, were found to have increased beyond those in 2019, as part of a generalized upward trend for child and adolescent admissions compared with the previous year. Individuals with ASD demonstrate a number of unique psychological and physical vulnerabilities which put them at greater risk during the pandemic. The loss of community services, the switch to remote learning, the pause in behavioural therapies, and the limitations in vocational opportunities represent some of the major ways in which this population was affected [43]. While large-scale studies on the impact of the pandemic in this population have been limited, the available evidence suggests that individuals with ASD were more likely to experience mood and anxiety disorders [44,45] and that in the first year of COVID-19 over half of children with ASD developed new psychiatric symptoms, such as irritability, sleep problems, and disruptive behaviour [45].

Further research into the long-term mental health consequences of the pandemic is a pressing area of need [46,47]. Proposals in this area include exploration of the chronic and direct effects of the virus itself, as well as the ways in which pandemic policy measures have impacted everyday life, especially with regards to vulnerable subgroups, which in addition to children and adolescents include the elderly, those from lower-income areas, and minority ethnic groups. Post-pandemic recommendations focus on an integrated healthcare approach encompassing equality of access to evidence-based care, such as telepsychiatry, mental health screening of COVID-19 survivors, and a consolidated community-care strategy encouraging improved liaison between primary care clinics, educational institutions, and workplaces. Recommendations also include the promotion of wellness through tried and tested strategies such as diet, exercise, mindfulness, and similar low-cost psychological approaches, along with a greater emphasis on suicide awareness and prevention.

Preliminary research into the long-term mental health sequelae of COVID-19 survivors points towards outcomes in terms of anxiety, depression, and post-traumatic stress disorder (PTSD) that are commensurate with the general population, further implicating the psychosocial, occupational and financial factors associated with the virus [48]. Conversely, there continues to emerge growing evidence on the long-term cognitive effects of COVID-19, which are mediated through the above-mentioned inflammatory pathways as well as hypoxia, and by means of direct binding to the angiotensin-converting enzyme 2 (ACE2), which is a prominent pathophysiological entry-point shared by respiratory epithelial cells and the olfactory nerve [39,40,49,50]. Long-term follow-up of COVID-19 patients has demonstrated persistent impairments in varying domains of cognitive and executive function [49,51], which is also in keeping with radiographic findings of frontotemporal hypoperfusion in these patients [52].

Our research was not without its limitations. While the broader elements of our region’s public health policies were similar to those implemented elsewhere, the variation across the world in responses to the pandemic may be associated with similarly varied mental health outcomes. Another limitation is associated with the appraisal of the results in populations for whom poorer mental health outcomes may not necessarily be reflected by an increase in hospital admissions, such as the very old or the very young. A further limitation of this study is that we did not delineate new admissions from readmissions, and therefore did not distinguish what proportion of the observed deterioration in outcomes was a product of decompensated illness in patients with known psychiatric histories, as opposed to de novo diagnosis. Finally, it is possible that personality disorders may have been underrepresented in our sample, and frequently contributed as an underacknowledged comorbidity in patients for whom the focus of the recorded discharge summary might have been a mood or substance-use disorder, for example. Therefore, the upward trend observed in 2021 might not reflect the true toll of the pandemic on this cohort.

## 5. Conclusions

We believe that the long-term increase observed in psychiatric inpatient admissions across the pandemic was a result of three major factors of varying significance. These were: (1) the direct pathogenic effects of the virus itself; (2) the secondary effects of COVID-19 on psychological, social, occupational, and economic wellbeing; and (3) delay in presentations to healthcare services, mediated by a combination of widespread anxiety and barriers to accessing care. The available evidence and the conclusions drawn from the correlational analysis in our study suggest that the latter two areas probably exerted a greater significance on the study results than the former. In light of this, we propose that the detrimental psychological impacts of any pandemic event be closely borne in mind in the guidance for any future policy planning, and that access to mental healthcare services be given high priority.

## Figures and Tables

**Figure 1 ijerph-20-01194-f001:**
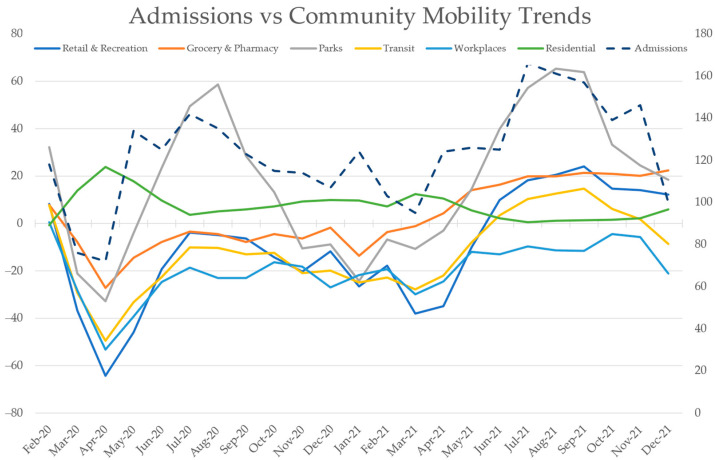
Total monthly psychiatric admissions (right axis, represented with the dashed line) charted against mean monthly community mobility (left axis, measured in percentage changes from baseline).

**Figure 2 ijerph-20-01194-f002:**
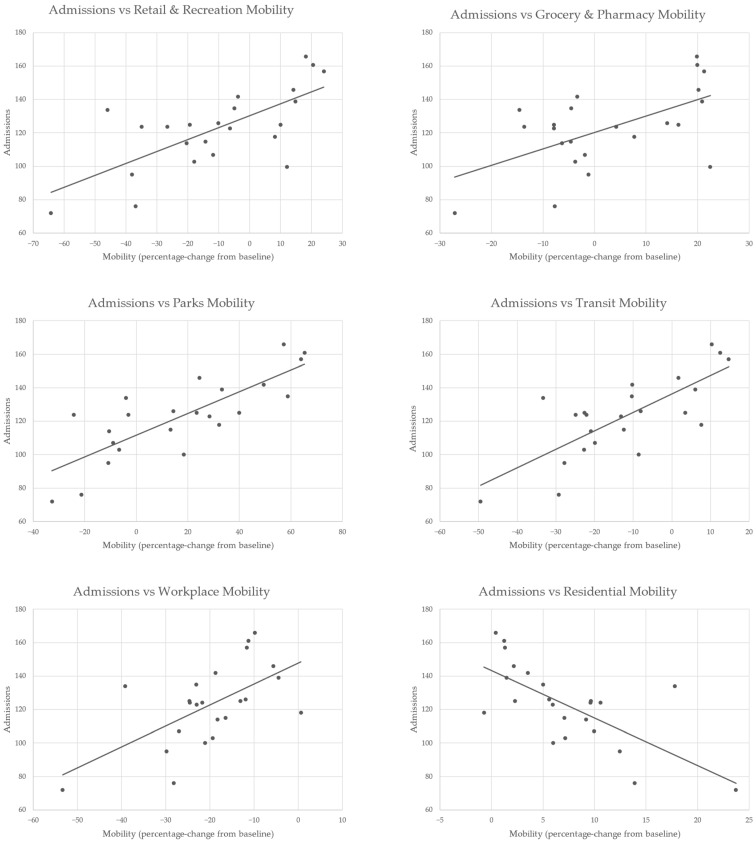
Scatter plots with lines of best fit for total monthly admissions vs. Community Mobility Reports for Malta.

**Table 1 ijerph-20-01194-t001:** Psychiatric Diagnosis: Comparison of mean monthly presentations for the first and second years of the pandemic vs. the pre-pandemic reference year.

	Mean Monthly Presentations			Difference			*p*-Value		
Diagnostic Group	2019 (SD)	2020 (SD)	2021 (SD)	2019 vs. 2020 (95% CI)	2019 vs. 2021 (95% CI)	2020 vs. 2021 (95% CI)	2019 vs. 2020	2019 vs. 2021	2020 vs. 2021
Organic disorders	2.6 (1.62)	2.9 (1.93)	4.3 (3.31)	0.3 (−1.16, 1.83)	1.7 (−0.47, 3.97)	1.4 (−1.35, 4.19)	0.630	0.110	0.284
Substance-use disorders	53.8 (8.24)	47.7 (12.47)	52.6 (8.99)	−6.1 (−12.74, 0.41)	−1.2 (−6.79, 4.29)	4.9 (−3.49, 13.33)	0.063	0.629	0.225
Schizophrenia, schizotypal and delusional disorders	17.9 (4.36)	20.1 (5.14)	23.8 (5.20)	2.2 (−2.87, 7.21)	5.9 (2.80, 9.04)	3.7 (−1.17, 8.67)	0.364	0.002	0.121
Mood disorders	27.2 (4.59)	33.8 (13.2)	42.5 (7.24)	6.6 (−1.87, 15.21)	15.3 (9.83, 20.84)	8.7 (2.88, 17.04)	0.114	<0.001	0.044
Anxiety disorders	15.8 (3.59)	15.0 (5.59)	20.3 (4.70)	−0.8 (−4.14, 2.47)	4.5 (1.24, 7.76)	5.3 (2.02, 8.64)	0.590	0.011	0.005
Eating disorders	0.1 (0.29)	0.3 (0.62)	0.8 (1.42)	0.2 (−0.2, 0.53)	0.7 (−0.29, 1.62)	0.5 (−0.50, 1.50)	0.339	0.151	0.293
Personality disorders	6.9 (2.02)	7.7 (5.16)	12.6 (6.50)	0.8 (−3.38, 4.88)	5.7 (1.52, 9.81)	4.9 (−1.14, 10.98)	0.697	0.012	0.102
Intellectual disability	3.0 (1.48)	2.7 (1.72)	4.3 (2.57)	−0.3 (−1.97, 1.30)	1.3 (−0.53, 3.20)	1.6 (−0.20, 3.53)	0.662	0.144	0.075
Autism spectrum disorder	1.5 (1.38)	1.8 (1.03)	3.0 (1.86)	0.3 (−0.73, 1.39)	1.5 (0.16, 2.84)	1.2 (−0.10, 2.43)	0.500	0.032	0.067
Childhood behavioural and emotional disorders	2.8 (1.64)	0.8 (0.97)	3.0 (2.26)	−2.0 (−3.18, −0.98)	0.2 (1.84, 2.17)	2.2 (0.71, 3.79)	0.002	0.858	0.008

**Table 2 ijerph-20-01194-t002:** Comparison of mean monthly presentations for age group, sex, nationality, and admission status.

	Mean Monthly Presentations			Difference			*p*-Value		
Age Group	2019 (SD)	2020 (SD)	2021 (SD)	2019 vs. 2020 (95% CI)	2019 vs. 2021 (95% CI)	2020 vs. 2021 (95% CI)	2019 vs. 2020	2019 vs. 2021	2020 vs. 2021
0–17	5.4 (1.44)	3.3 (1.97)	5.8 (3.31)	−2.1 (−3.53, −0.64)	0.4 (−2.26, 0.87)	2.5 (−0.01, 4.85)	0.009	0.710	0.051
18–29	29.8 (5.31)	32.5 (8.94)	31 (7.90)	2.7 (−3.65, 8.98)	1.2 (−3.05, 5.38)	−1.5 (−6.83, 3.83)	0.373	0.555	0.548
30–39	26.7 (5.12)	30.3 (6.92)	33.5 (8.42)	3.6 (−0.86, 8.03)	6.8 (2.19, 11.48)	3.2 (−2.76, 9.26)	0.104	0.008	0.259
40–49	22.2 (3.16)	21.3 (7.20)	22.9 (4.64)	−0.9 (−5.59, 3.92)	0.7 (−3.01, 4.51)	1.6 (−3.07, 6.23)	0.707	0.669	0.469
50–59	15.3 (3.77)	14.4 (4.14)	18 (3.79)	−0.9 (−4.34, 2.50)	2.7 (−1.25, 6.59)	3.6 (0.25, 6.92)	0.567	0.162	0.037
60–69	7.4 (2.47)	7.8 (2.69)	13 (4.69)	0.4 (−1.22, 2.05)	5.6 (2.15, 9.02)	5.2 (1.51, 8.82)	0.586	0.004	0.01
70+	5.5 (2.75)	5.1 (2.94)	6.3 (2.71)	−0.4 (−2.91, 2.07)	0.8 (−2.06, 3.72)	1.3 (−0.99, 3.49)	0.720	0.539	0.244
Sex									
Male	76.7 (7.44)	79.7 (18.30)	80.7 (10.55)	3.0 (−7.35, 13.35)	4.0 (−1.78, 9.78)	1 (−8.09, 10.09)	0.537	0.156	0.813
Female	35.7 (7.01)	35.2 (6.60)	49.8 (15.46)	−0.5 (−4.37, 3.37)	14.1 (6.29, 22.05)	14.6 (5.53, 23.80)	0.781	0.002	0.005
Nationality									
Maltese	89.7 (9.85)	86.3 (17.20)	102.8 (19.17)	−3.4 (−12.14, 5.31)	13.1 (1.74, 24.43)	16.5 (3.72, 29.28)	0.407	0.028	0.016
Non-Maltese	22.7 (5.02)	28.7 (6.89)	27.8 (7.21)	6.0 (2.17, 9.83)	5.1 (−0.05, 10.21)	−0.9 (−5.46, 3.63)	0.005	0.052	0.666
Admission Status									
Voluntary	80.8 (8.96)	78.0 (17.52)	84.1 (17.33)	−2.8 (−12.80, 7.13)	3.3 (−5.63, 12.13)	6.1 (−5.51, 17.68)	0.544	0.437	0.273
Involuntary	31.5 (6.57)	36.8 (11.33)	46.4 (9.01)	5.3 (−4.55, 15.22)	14.9 (7.37, 22.46)	9.6 (4.05, 15.12)	0.260	0.001	0.003

**Table 3 ijerph-20-01194-t003:** Comparison of mean monthly presentations with non-suicidal self-injury (NSSI), suicidal ideation (SI), and suicidal self-injury (SSI).

	Mean Monthly Presentations			Difference			*p*-Value		
	2019 (SD)	2020 (SD)	2021 (SD)	2019 vs. 2020 (95% CI)	2019 vs. 2021 (95% CI)	2020 vs. 2021 (95% CI)	2019 vs. 2020	2019 vs. 2021	2020 vs. 2021
NSSI	5.6 (2.54)	8.6 (7.33)	9.0 (4.69)	3 (−1.76, 7.76)	3.4 (0.41, 6.43)	0.4 (−4.83, 5.67)	0.193	0.030	0.864
SI	23.2 (3.79)	36.1 (13.85)	44.8 (11.26)	12.9 (5.21, 20.62)	21.6 (15.34, 27.83)	8.7 (2.49, 14.85)	0.004	<0.001	0.01
SSI	7.8 (2.08)	11.3 (5.19)	13.3 (4.19)	3.5 (0.01, 6.83)	5.5 (2.38, 8.62)	2 (−1.15, 5.32)	0.050	0.003	0.184

**Table 4 ijerph-20-01194-t004:** Summary of Pearson’s correlation of monthly psychiatric admissions with mean monthly community mobility data (units represent percentage change from baseline).

	No.	Mean	SD	Min	Max	r (with Admissions)	*p*-Value
Admissions	23	122.9	24.15	72	166		
Retail & Recreation	23	−10.2	23.64	−64.3	24.0	0.699	<0.001
Grocery & Pharmacy	23	2.7	14.13	−27.1	22.5	0.577	0.004
Parks	23	17.3	29.75	−32.8	65.3	0.800	<0.001
Transit	23	−12.1	16.56	−49.5	14.7	0.755	<0.001
Workplaces	23	−19.8	11.69	−53.3	0.6	0.607	0.002
Residences	23	7.2	5.95	−0.7	23.7	−0.699	<0.001

## Data Availability

The datasets generated and/or analysed during the current study are not publicly available because the original data contain confidential patient information. The datasets generated are available from the corresponding author on reasonable request. This does not apply to any datasets containing information that could identify individual patients.

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
