# Peer review of "A Retrospective Longitudinal Analysis of Mental Health Admissions: Measuring the Fallout of the Pandemic"

_ijerph, 2023, doi:10.3390/ijerph20021194_

Round 1
Reviewer 1 Report
this paper is reviewing the changes in admissions to the mental health hospital in Malta during the two first pandemic years of COVID19 pandemic. the study is based on the database of admissions and is thus quantitative in nature.
I feel based on the richness of the data that formed the base of this study, the study self presents a very limited picture of the impact of the pandemic on hospital admissions. The conclusion of the study is quite simply put: the pandemic had a negative effect on the rate of admissions. But this is not new information.
It would be more interesting to see in detail why these changes appeared and may be link them to the major events Malta's response to the pandemic - make a timeline of major lock-downs, policies and campaigns and see what happened with admissions around these time points. Introducing this approach would mean also restructuring the introduction part of the paper
Additionally, I would recommend also running an analysis of 2020 vs 2021. Why was it left out of the analysis?
All in all, I think the paper as it is now, is quite well written and the stats are done and presented in an adequate manner, my concern is about the added value of the findings.
Author Response
To whom it may concern
We sincerely appreciate you finding the time to review our paper. We've taken your concerns on board, and have done our best to address them by diving deeper into the data, running further analyses, and giving a more in-depth overview of the pandemic timeline in Malta.
I'll go through our amendments in the order in which we've addressed them in the paper.
- Background
- Timeline of major lockdowns
We've provided a more in-depth review of Malta's pandemic-timeline and the local public health policies which followed in the background section of our manuscript. We've expanded this subject to a dedicated overview of around 500 words in the background section.
-
- Reviewing the relationship between admissions and pandemic timeline
Furthermore, we have carried out a secondary analysis with the aim of tying the pandemic timeline with hospitalizations. We did this using the Community Mobility Reports available from Google which we made reference to in the discussion in the original paper. These are country-by-country measures of movement of populations over time and across 6 geographical domains (such as workplaces, parks and residences). We believe these reports provide a fascinating and objective parameter against which we could correlate our data. We introduce the mobility reports in the background of our paper.
- Methods
- Reviewing the relationship between admissions and pandemic timeline
We took the monthly psychiatric hospitalization totals and correlated them with the mean monthly mobility data from Google, which is publicly available for research purposes, using Pearson's correlation.
This data was first shared in February 2020, and was available until the end of 2021, which is the end of our study period. We were therefore able to correlate admissions with mobility for a 23 month period.
-
- 2020 vs 2021 analysis
We ran an analysis of 2020 vs 2021 for all of the areas we explored in the original manuscript. The only reason it wasn't included in the original manuscript is that we wanted to compare each year of the pandemic with a "control" or "reference" year, which we took as 2019.
- Results
- Reviewing the relationship between admissions and pandemic timeline
We've presented the results of our second analysis in two additional figures and one additional table. Figure 1 explores the mobility data from Google and how this changed over time for our study period. We've superimposed the monthly admission totals on the chart to provide a visual representation of how hospitalizations changed with the various mobility metrics which reflect COVID-19 restrictions and public attitudes towards the virus in Malta.
Figure 2 shows the scatter plots of monthly admissions vs each of the mobility metrics and the results of the Pearson's correlation are presented in Table 4. We observed varying degrees of significant correlation, the majority of which were strong to very strong.
-
- 2020 vs 2021 analysis
All of the results of our separate 2020 vs 2021 analysis have been included in Tables 1 to 3 and we've made several new references to significant results in the text. Furthermore, we've tried to make the data presented in the tables clearer with shading, borders, reduced font sizes and increased row height.
- Discussion
- Reviewing the relationship between admissions and pandemic timeline
Once again, we felt that the mobility reports represented an interesting and objective way of measuring the product of various factors of the pandemic on the Maltese population. The results of the correlation strongly suggest that psychiatric presentations were delayed as a result of these factors. Furthermore, we believe that this analysis is a unique one in the literature covering the impact of the pandemic on mental health.
In the discussion, we explore the mobility reports and how they tie in with hospital admissions within the broader context of the pandemic timeline in Malta.
___END OF REVIEW___
Once again, thank you for finding the time to review our paper. We hope that we were able to address your concerns.
Regards
Dr Sean Warwicker
Reviewer 2 Report
The authors have presented their research with clarity and plausability. The Abstract is concise and presents the project. The Introduction provides an excellent explanation for the research project which addresses the mental health issues resulting from social isolation etc. imposed by the COVID-19 epidemic. The research has a sound methodological base for the collection of qualitative and quantitative data which has been analyzed using appropriate statistical methods while the qualitative provides the phenomenological narrative data to support the research. The statistical and narrative data are interpreted in a clear and concise manner in the Discussion and the Conclusion. The use of medical data to clarify the development of the psychological behaviors of people offers an insight into how a person's mind can be distorted by isolation, and the long-term effects of these mental disorders.
This is an important research paper into the effects of pandemics and isolation of people to control the pandemic. The authors are commended for their foresight and diligence in undertaking the appropriate research to analyze an observed social problem which can only be researched retrospectively.
Suggested alterations to the text are shown in the attached file.

Author Response
To whom it may concern
First off, we genuinely appreciate your positive feedback. It was a pleasure to read your responses to our manuscript. We've carried out all of the alterations to the text which you have suggested, with some minor changes to suit the flow of language.
- "now that the dust of lockdowns and other related restrictive
healthcare strategies has settled" has been changed to "now that the consequences of lockdowns and other related restrictive healthcare strategies have become established." - "once the virus took root" has been changed to "As viral cases only continued to rise". This portion of the text has changed as another reviewer proposed a more in-depth analysis of the COVID-19 timeline in Malta in the introduction of our paper.
- The two sentences starting with numerals have been corrected. They now read: "During this 3-year period, 5,760 patients presented ..." and "A total of 4,292 psychiatric admissions were therefore recorded ...", respectively.
- "when one’s attention shifts from the trees to the wood" has been changed to "when one takes a broader perspective."
- " a battery of" has been changed to "a considerable amount of".
- The second "that" has been removed".
___END OF REVIEW__
There have been broader changes in the text, as one reviewer had proposed that we provide a more in-depth review of the COVID-19 timeline in Malta, that we show the results of a 2020 vs 2021 analysis, and that we dive a bit more into our data to improve the richness of the paper. In order to meet this, we looked into our findings correlated with Community Mobility Reports from Google, which are measures of population movement over time and different geographical domains. We believe this gave us some fascinating results.
Once again, thank you for your kind review, and for taking the time to review our paper.
Regards
Dr Sean Warwicker
Reviewer 3 Report
Thank you for counting on me for this review. Congratulations on your choice of an interesting and topical subject. In the attached document you have the changes I suggested after reviewing your work.
Best regards.

Author Response
To whom it may concern
Thank you for finding the time to review our paper. We sincerely appreciate the positive feedback and we've done our best to take your feedback on board and amend our manuscript. I'll go through the points raised below:
- Abstract
- Numbering brackets
The numbering brackets were a what we had used to numerically order the sequence of the abstract (for example, (1) Background ... (2) Methods). We have removed these for the sake of clarity and ensured that there are no bibliographical references or citations in the text here.
-
- Reference to psychiatric centre
We've used the name of our hospital where the research was carried out, which is Mount Carmel Hospital in Malta.
-
- Avoiding acronyms
All acronyms have been removed and/or replaced in our abstract.
- Materials and methods
- Use of the term "control year"
All use of the term "control year" has been substituted with the term "reference year".
-
- Inclusion and exclusion criteria
We've reviewed this part of our paper and lead this section with a specific introductory sentence on what our inclusion criteria were (all patients who were either newly admitted or readmitted to Mount Carmel Hospital for a primary psychiatric indication) to ensure greater clarity. We've been clearer in the text in detailing each of the excluded groups. Furthermore, to avoid duplicating information as you rightly point out, we've removed the flow chart, as all information it presents is now clearly documented in the text.
- Results
- Avoiding the duplication of information
We've reviewed this portion of our paper and deleted numerous instances where values were mentioned both in the text and in the tables. The introductory portion to our results (lines 105-108), now reads as follows:
"The mean monthly admission data regarding clinical diagnosis is laid out in Table 1. The highest admission rates in 2019 were for substance use disorders; mood disorders; schizophrenia, schizotypal and delusional disorders; anxiety disorders; and personality disorders, respectively. As can be observed in the demographic data recorded in Table 2, voluntary admissions exceeded involuntary and males were much more likely to be admitted than females during the year. "
We've ensured that the only references to the tables which then follow in the text are to highlight important statistically significant changes, such as the increase in involuntary admissions, or the increase in patients who presented year-on-year with suicidal ideations, or suicidal self-injury. Another reason we wanted to ensure these were present was because another reviewer proposed that we include the results of a 2020 vs 2021 analysis in our manuscript, so we wanted to ensure the reader would be able to draw the most important conclusions from the text.
-
- Tables
Thank you for pointing this issue out. We've used a combination of shading, borders, reduced font sizes and increased row height to ensure our information is presented as clearly as possible. Again, it was proposed that we add the results of a comparison of 2020 to 2021, so we've done our best to present all of this information in individual tables.
-
- Units of data
We've ensured that all references to the results in our tables have the appropriate units.
For example: "The total number of admissions for the year was similar to 2019, although an increase in nonMaltese admissions may 122 have reflected a disproportionate socio-economic impact of the
pandemic (22.7 vs. 28.7, 123 p=0.005)." now reads: "The total number of admissions for the year was similar to 2019, though an increase in non-Maltese admissions may have reflected a disproportionate socio-economic impact of the pandemic (22.7/month vs 28.7/month, p=0.005)."
___END OF REVIEW___
Again, thank you for finding the time to review our paper, and we hope that we've adequately addressed all of your concerns.
Regards
Dr Sean Warwicker
Round 2
Reviewer 3 Report
Dear authors:
For my part I consider that no further changes are necessary, I agree with your answers. I would like to point out that the inclusion of the comparison between 2020 and 2021 was a very good move on your part.
Finally, I would like to congratulate you once again on your choice and approach to the topic of study.
Best regards.